# Germination, Physicochemical Properties, and Antioxidant Enzyme Activities in Kangkong (*Ipomoea aquatica* Forssk.) Seeds as Affected by Dielectric Barrier Discharge Plasma

Prapasiri Ongrak [1], Nopporn Poolyarat [2], Suebsak Suksaengpanomrung [2], Kamtorn Saidarasamoot [2], Yaowapha Jirakiattikul [3] and Panumart Rithichai [3,*]

[1] Department of Biotechnology, Faculty of Science and Technology, Thammasat University, Pathum Thani 12120, Thailand; prapasiri1994@gmail.com
[2] Center of Advanced Nuclear Technology, Thailand Institute of Nuclear Technology (Public Organization), Nakhon Nayok 26120, Thailand; noppornp@tint.or.th (N.P.); suebsak@tint.or.th (S.S.); kamtorn@tint.or.th (K.S.)
[3] Department of Agricultural Technology, Faculty of Science and Technology, Thammasat University, Pathum Thani 12120, Thailand; yjirakia@tu.ac.th
* Correspondence: panumart@tu.ac.th; Tel.: +66-2564-4525

**Abstract:** Dielectric barrier discharge (DBD) plasma has been utilized as a sustainable technology to enhance seed germination in various plant species. The objective of this research was to identify the mechanism of physicochemical properties and antioxidant enzyme activities to promote kangkong (*Ipomoea aquatica* Forssk.) seed germination using different durations of DBD plasma treatments. Seeds were exposed to atmospheric DBD plasma from 5 to 20 min, compared to non-treated seeds as the control. According to SEM images, the seed surface had cracks and grew wider as a result of the prolonged DBD plasma treatments. A longer DBD plasma treatment exhibited a lower water contact angle and increased water absorption. DBD plasma treatments strongly improved germination percentages and hydrogen peroxide ($H_2O_2$) contents. Seeds treated with DBD plasma for 20 min showed the highest malondialdehyde (MDA) content and the lowest field emergence. Catalase (CAT) activity increased under DBD plasma treatments for 5 and 10 min. Ascorbate peroxidase (APX) and superoxide dismutase (SOD) activities were not statistically different among the treatments. This finding suggested that DBD plasma treatments stimulated the germination of kangkong seeds by modifying the seed surface, and upregulating $H_2O_2$ content and CAT activity. Five minutes was an appropriate time to treat DBD plasma.

**Keywords:** catalase; DBD plasma; hydrogen peroxide; *Ipomoea aquatica* Forssk.; malondialdehyde; wettability

## 1. Introduction

Kangkong (*Ipomoea aquatica* Forssk.) belongs to the family Convolvulaceae and is naturally grown in tropical regions of Africa, Asia, and India [1,2]. It is one of the most commonly consumed green vegetables in Thailand. Kangkong has been reported as a valuable source of secondary metabolites, such as flavonoids, alkaloids, vitamins, and minerals [3]. It possesses medicinal properties, including anti-inflammatory, hypolipidemic, antioxidant, anti-glycation, anti-diabetic, and α-glucosidase inhibitory effects [4,5]. Kangkong is propagated by seeds and is typically grown for both microgreens and leafy greens, requiring high-quality seeds to promote rapid germination and uniform seedling growth. Nevertheless, kangkong seeds faced difficulty in germination and achieving good field establishment due to seed coat-imposed dormancy [6]. For kangkong cultivation, conventional seed pretreatments, such as water soaking, heat treatment, or scarification, are commonly employed to achieve high seed germination rates and ensure uniform seedling

establishment. According to Ebert and Wu [6], the germination rate of kangkong seeds, which is initially below 60%, can be enhanced by partially removing the seed coat, followed by a 24 h soaking period in water. This method results in an increased germination rate of over 80%. However, these processes can potentially damage the seeds, be time consuming, and generate waste. Therefore, there have been numerous efforts to identify effective and sustainable methods for enhancing seed quality.

Plasma technology is considered a novel and environmentally friendly method. Dielectric barrier discharge (DBD) plasma is one of the most popular non-thermal plasma techniques. It is generated by accumulating and discharging ions on a dielectric surface using alternating current electricity at a high electric potential difference in atmospheric pressure or other gaseous mediums [7]. DBD plasma treatment provided reactive species, electrons, positive and negative ions, ultraviolet (UV) radiation, and heat [8]. These are effective elicitors that can promote seed germination, plant growth development, and metabolism. Plasma treatment has been recognized as a promising and effective technology for enhancing the germination of various plant species [9–19]. However, the effect of plasma treatment on kangkong seed germination has not been reported. The possible effects of non-thermal plasma on the various aspects of plants depend on plasma conditions and treatment times [9]. For example, the germination rate of black gram seeds treated with DBD plasma treatment for 120 s increased while, for 180 s, it decreased [10]. According to Zahoranová et al. [11], wheat seed subjected to 30 s of plasma treatment exhibited the highest percentage of germination and vigor index, while longer exposure times had an adverse effect. Similarly, cucumber seeds treated with DBD plasma at a high voltage of 25 kV for 10 s showed higher germination and growth than the control, but the prolonged DBD plasma treatment of 50 s had a negative impact on seed germination and seedling growth [12]. After a 5 min rollable DBD plasma treatment, the germination rate of spinach seeds increased by 15% compared to the control. Conversely, treatments lasting 1 and 3 min did not exhibit significant differences from the control [13]. Exposure of rice seeds to DBD plasma for durations of 2, 5, 10, 15, and 20 min exhibited a positive impact on germination. Among these durations, seeds treated with DBD plasma for 20 min displayed the highest values in germination potential, germination rate, and germination index, showing increases of 36.73%, 26.00%, and 25.92%, respectively, compared to untreated seeds [14]. Improving seed germination by plasma treatment involves numerous mechanisms, including seed coat modification, biochemical changes, and antioxidant enzyme activity. Seed surface modification after plasma treatment improving seed germination has been reported in various species such as spinach [15], rice [16], artichoke [17], black gram [10], and radish [18]. The changes in the seed coat surface caused by plasma treatment led to changes in the wettability of the seed [13,18]. The increase in surface wettability of the treated seeds induced higher water uptake, resulting in accelerated seed germination [19]. The water uptake of black gram seed after plasma treatment at 20 to 180 s increased and became faster with the increase in plasma treatment time [10].

During plasma treatments, reactive oxygen species (ROS) is typically generated. Crucial ROS, specifically the superoxide anion ($O_2^{\bullet-}$), hydroxyl radical ($\bullet OH$), and hydrogen peroxide ($H_2O_2$), are recognized as the major agents to triggering oxidative stress. These ROS cause abiotic stress, encouraging the activation of biochemical pathways and the generation of various specialized antioxidant compounds within the plant [20]. The intracellular oxidative stress was regulated by ROS production and the intracellular defense mechanisms [21]. Among these ROS, $H_2O_2$ is a long-lived reactive molecule that plays a dual role in plant physiology and developmental processes [22]. It can interact with the seed coat and diffuse through the membrane [8], activating specific metabolic pathways after imbibition [23]. The increase in $H_2O_2$ content following plasma treatment has been observed to stimulate seed germination [24,25]. Moreover, highly active and unstable ROS affected molecular biology in plant cells, leading to oxidative stress damage to lipids, proteins, and amino acids causing cell damage, and cell membrane deterioration including lipid peroxidation [22,26]. Malondialdehyde (MDA) has been used as a direct indicator of

membrane deterioration and lipid peroxidation [27]. Wheat seed exposed to DBD plasma for 30 s and 180 s showed a significant increase in MDA levels of 27.8% and 33.8%, respectively, higher than the control [28]. Plants have stress defense through phytochemical and antioxidant systems, which are used in the detoxification of the ROS causing oxidative stress [26]. Antioxidant enzymes, including superoxide dismutase (SOD), ascorbate peroxidase (APX), and catalase (CAT) are important enzymatic components [29]. These enzymes eliminate excess oxidants while protecting plant cells [30]. Changes in antioxidant enzyme activities related to enhanced seed germination have been reported. The elevation in SOD and CAT activities following a 15 s exposure to non-thermal plasma treatment promoted the germination of sunflower seeds [30]. Similar results were also reported by Cui et al. [25], who demonstrated that short-term plasma treatment ($\leq$3 min) elevated CAT, SOD, and peroxidase activities, resulting in improved germination in *Arabidopsis* seeds. Based on this information, mechanisms to control seed germination vary among plant species. However, the mechanism to stimulate kangkong seed germination using DBD plasma was still limited. Therefore, the objective of this study was to investigate the effect of different durations of DBD plasma treatment on kangkong seed germination and elucidate the regulatory effects of DBD plasma on seed surface and biochemical properties, as well as antioxidant enzyme activities in seeds. Understanding the mechanism of plasma-treated seeds will help in establishing regulations for this technology.

## 2. Materials and Methods

### 2.1. Seed Sample, Experimental Setup, and DBD Plasma Treatment

Kangkong seed, cv. Pugun 19, was used in this study. It was purchased from Home Seeds Corporation Co., Ltd., Pathum Thani, Thailand.

DBD plasma was generated at atmospheric pressure using a DBD device consisting of two electrodes, a dielectric layer, and a power supply as shown in Figure 1. The bottom electrode was a copper plate with a dimension of $10 \times 10 \times 0.2$ cm$^3$, while the top electrode was a liquid, normal saline, contained in the glass cabinet, in which act as dielectric material. The gap between the bottom electrode and the glass cabinet was set to 6 mm. To generate DBD plasma, the HV power supply (Trek 30/20A) was operated at a voltage of 5.5 kV, with a frequency of 4.0 kHz. The monitoring capacitor was inserted in the circuit to help evaluate the discharge power. By measuring voltage across the monitoring capacitor, ones can obtain the current in circuit. Together with measuring the voltage across the DBD gap, the discharge power can be calculated from

$$\text{Discharge power} = [\text{C}_{\text{monitoring}} \times \text{V}_c] \times \text{V}_{\text{gap}}$$

where $\text{C}_{\text{monitoring}}$, $\text{V}_c$, and $\text{V}_{\text{gap}}$ represented capacitance of monitoring capacitor, voltage across monitoring capacitor, and voltage across DBD gap, respectively.

For our DBD apparatus, the average discharge power was found to be 59.7 W.

For DBD plasma treatment, three hundred seeds were spread in a single layer on a copper plate ($10 \times 10$ cm$^2$) in the generator. The seeds were treated with atmospheric DBD plasma for 5, 10, 15, and 20 min. The non-treated seeds were used as a control.

### 2.2. Seed Surface Morphology

Dried DBD plasma-treated seeds and non-treated seeds were randomly selected to determine the surface morphology. The seeds were coated with titanium before being placed in the chamber and imaged using a scanning electron microscope (SEM; Hitachi SU5000, KINO Scientific Instrument Inc., Boston, MA, USA).

### 2.3. Wettability Test

To determine the surface wettability, the water contact angle was measured. A droplet of deionized (DI) water, 10 μL, was placed onto the DBD plasma-treated and non-treated seed surfaces. The images were taken by a camera coupled to a computer and recorded using a contact angle meter and contact angle goniometer from the SL250 series.

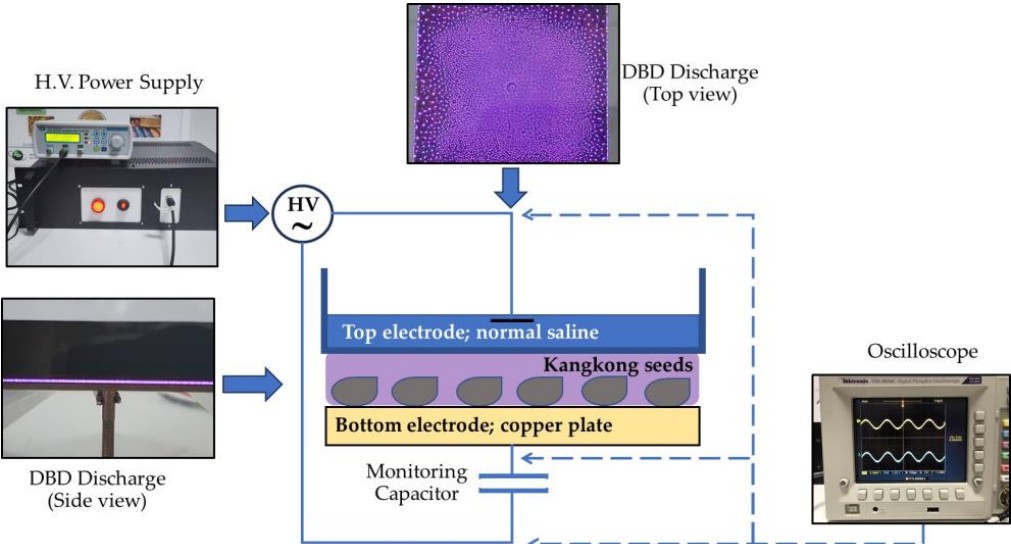

**Figure 1.** Diagram of DBD apparatus and experimental setup.

*2.4. Water Absorption*

Forty seeds were weighed and placed in a Petri dish containing 25 mL of distilled water. They were kept at 25 °C. To determine water absorption, the seeds were weighed after soaking for 3, 6, 9, 12, 15, 18, and 21 h. Water absorption was determined using the following formula [9]:

$$\text{Water absorption (\%)} = [(W_w - W_d)/W_d] \times 100$$

where $W_d$ and $W_w$ represented the weights of the dry and wet seeds, respectively.

*2.5. Germination Test*

Fifty seeds were germinated using the between paper (BP) method as described by ISTA [31]. They were incubated in a germinator chamber at 30 °C, with light provided for 8 h, and watered daily. Seed germination was determined when the radicle protruded at least 2 mm, and counting was performed on day 10. The germination percentage was calculated using the following formula:

$$\text{Germination (\%)} = (\text{number of germinated seeds}/\text{number of total seeds}) \times 100$$

*2.6. Field Emergence*

Fifty seeds were sown in a $30 \times 45 \times 15$ cm$^3$ basket filled with moist peat moss. They were kept in a plastic house. Watering was completed daily. The number of germinated seeds was recorded when a hypocotyl length of at least 2 mm was visible. The field emergence was calculated using the same formula as the germination percentage.

*2.7. Hydrogen Peroxide ($H_2O_2$)*

$H_2O_2$ content was determined following the methods described by Billah et al. [10]. Briefly, 0.5 g of seed powder was mixed with 5 mL of 0.1% trichloroacetic acid (TCA). The mixtures were centrifuged at $10,000 \times g$ for 15 min. One mL of supernatants was then mixed with 2 mL of 1 M potassium iodide and 1 mL of 10 mM phosphate buffer (pH 7.0) and kept in the dark for 1 h. The absorbance was measured at 390 nm using a spectrophotometer (Shimadzu Europe UV-1208). The $H_2O_2$ content was calculated and expressed as mmol·g$^{-1}$ FW.

### 2.8. Malondialdehyde (MDA)

MDA content was measured using the thiobarbituric acid (TBA) reaction [32]. In brief, 200 mg of seed powder was mixed with 2 mL of 0.1% TCA and centrifuged at $12,000 \times g$ for 15 min. The supernatants were then combined with 1.5 mL of 20% TCA and 0.5% TBA, incubated in boiling water for 30 min, and quickly cooled in an ice bath. Afterward, the mixtures were centrifuged at 10,000 rpm for 10 min. The absorbance was measured at 532 and 600 nm, and the amount of MDA–TBA complex was determined using a molar extinction coefficient of 155 $mM^{-1} \cdot cm^{-1}$. The results were expressed as $nmol \cdot g^{-1}$ FW.

### 2.9. Electrical Conductivity (EC) Test

Twenty-five seeds were weighed and soaked in a beaker containing 75 mL of distilled water. They were kept at 25 °C for 24 h. After soaking, EC was determined using a conductivity meter (EC tester Hanna HI98311). The electrical conductivity value was calculated using the following formula [33]:

$$EC = (EC \text{ of sample} - EC \text{ of control})/\text{seed weight}$$

The EC value was expressed as a $\mu S \cdot cm^{-1} \cdot g^{-1}$.

### 2.10. Antioxidant Enzymes

For enzyme extraction, 0.2 g of seed powder was mixed with 0.5 mL of 50 mM sodium phosphate buffer (pH 7.0) and 0.4 μg of polyvinylpyrrolidone (PVP) in an ice bath. After centrifugation at $15,000 \times g$ for 15 min at 4 °C, the supernatant was used for measuring APX [34] and CAT [35] activities. For SOD, 0.5 g of seed powder was mixed with 0.5 mL of 100 mM sodium phosphate buffer (pH 7.8) and 0.4 μg of PVP before undergoing the same centrifugation process [36].

The activities of APX and SOD were determined using the methods described by Heshmati et al. [34]. The CAT activity was evaluated based on the methods provided by Önder et al. [35]. The antioxidant enzyme activities were expressed in $Unit \cdot mg^{-1} \cdot protein^{-1}$.

### 2.11. Statistical Analysis

The experimental design followed a completely randomized design (CRD) with five treatments and three replications. The data were analyzed using analysis of variance (ANOVA) with SPSS software version 21.0, and mean separation was performed using Tukey's honestly significant difference (HSD) test at a significance level of $p < 0.05$.

## 3. Results

### 3.1. Seed Surface Morphology

The surface of the kangkong seeds was altered after exposure to DBD plasma. The seed surface was damaged and broken after DBD plasma treatment. Additionally, as the exposure period increased, it gradually became rougher, and the cracks grew broader (Figure 2).

### 3.2. Wettability

DBD plasma treatments improved seed wettability. The water contact angle of DBD plasma-treated seeds was notably lower in comparison to the control. With a longer DBD plasma treatment, a lower water contact angle was obtained. The seeds treated with DBD plasma for 20 min exhibited the lowest water contact angle of 47.38 ± 2.26°, whereas the control showed the highest water contact angle of 82.93 ± 2.45° (Figures 3 and 4).

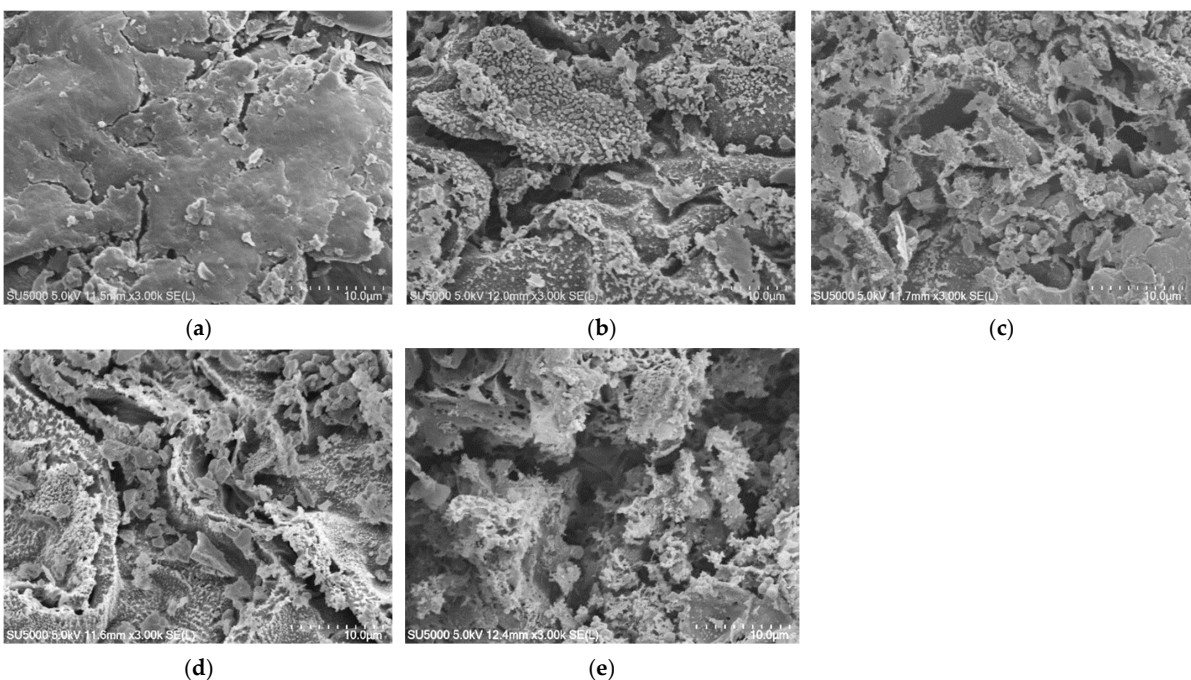

**Figure 2.** SEM images of the kangkong seed surfaces: non-treated seed (**a**); and DBD plasma-treated seed for 5 (**b**); 10 (**c**); 15 (**d**); and 20 min (**e**). The magnification for all images was 3000×.

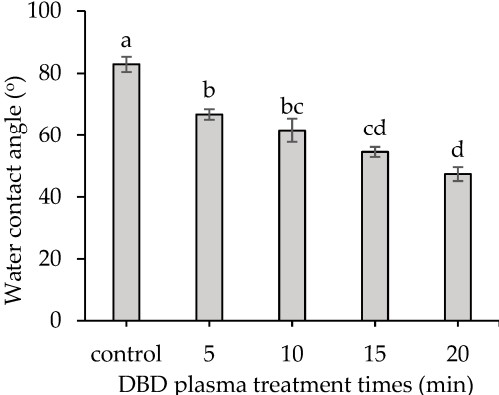

**Figure 3.** Effect of DBD plasma treatments on the water contact angle of the kangkong seed surface. The data represent means of three replicate samples, and error bars indicate ± SD, with the same lowercase letter indicating a non-significant difference based on Tukey's honestly significant difference test at $p < 0.05$.

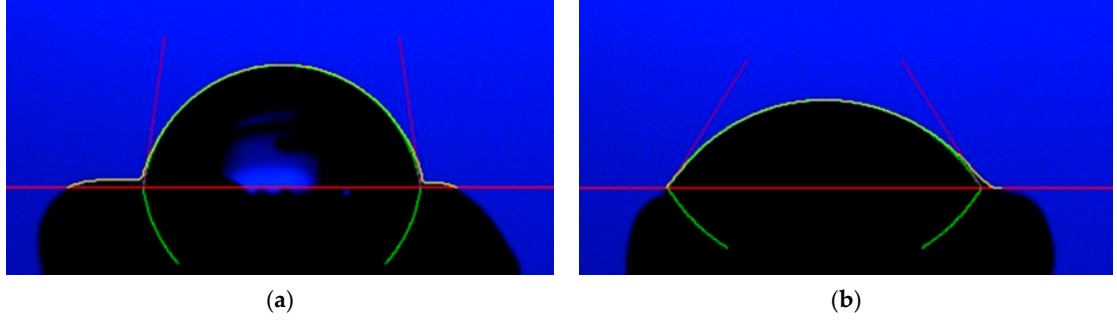

**Figure 4.** The water contact angle of the water drop on the surface of non-treated seed (**a**); and DBD plasma-treated seed for 20 min (**b**). The red, yellow, green, and black lines represent base, droplet, refection of droplet, and contact angle degree, respectively.

### 3.3. Water Absorption

DBD plasma-treated seeds showed faster water uptake compared to the control. It was observed that longer DBD plasma treatment times resulted in increased water absorption, regardless of imbibition times. The control exhibited a maximum water absorption of 16.96% at 21 h. In contrast, 20 min of DBD plasma treatment showed the highest water absorption of 62.11% at the same imbibition time (Figure 5).

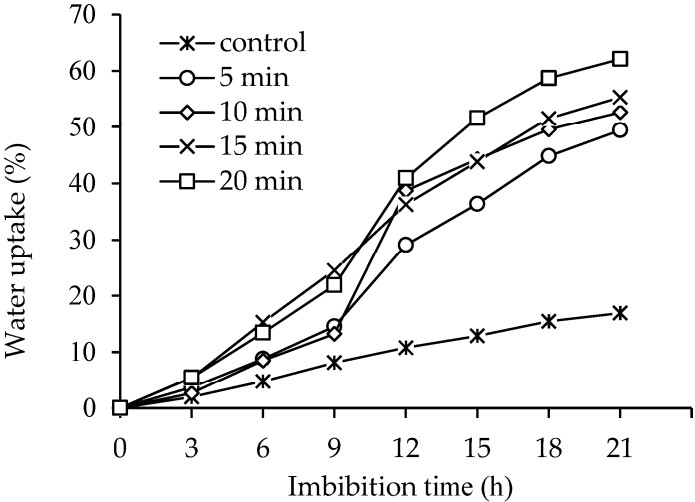

**Figure 5.** Effect of DBD plasma treatments on water uptake of kangkong seeds.

### 3.4. Seed Germination

The germination percentage was dramatically increased by DBD plasma treatments. The 15 min DBD plasma treatment demonstrated the highest germination percentage of $96.67 \pm 0.94\%$, which was not significantly different from the germination percentages of the 5-, 10-, and 20 min DBD plasma treatments (ranging from $94.67 \pm 1.89$ to $96.00 \pm 1.63\%$). The non-treated seeds revealed the lowest germination rate, which was $86.00 \pm 4.32\%$ (Figure 6a).

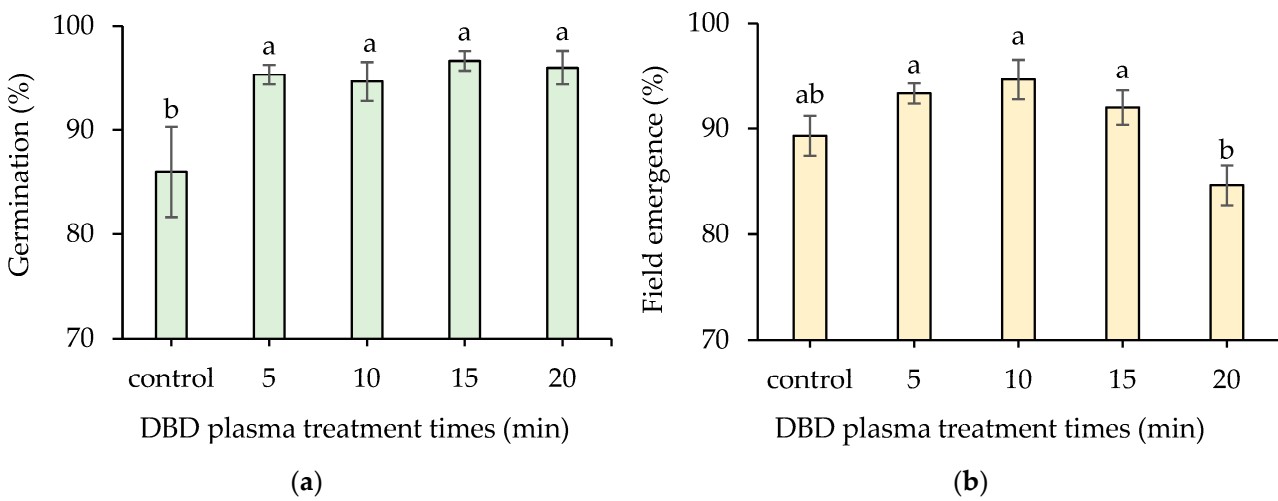

**Figure 6.** Effect of DBD plasma treatments on germination percentage (**a**); and field emergence (**b**) of kangkong seeds. The data represent means of three replicate samples, and error bars indicate $\pm$ SD, with the same lowercase letter indicating a non-significant difference based on Tukey's honestly significant difference test at $p < 0.05$.

### 3.5. Field Emergence

The highest field emergence of 94.67 ± 1.89% was obtained from a 10 min DBD plasma treatment, which was not significantly different from those of the 5 and 15 min DBD plasma treatments (93.33 ± 0.94% and 92.00 ± 1.62%, respectively). The lowest field emergence of 84.67 ± 1.89% was observed from a 20 min DBD plasma treatment (Figure 6b).

### 3.6. Hydrogen Peroxide

DBD plasma treatments statistically enhanced the content of $H_2O_2$, regardless of exposure times. The 20 min DBD plasma treatment showed the greatest $H_2O_2$ content of $82.54 \pm 3.36$ mmol·$g^{-1}$ FW and it was not significantly different from those of the 5, 10, and 15 min DBD plasma treatments (ranging from $79.59 \pm 0.90$ to $82.44 \pm 1.24$ mmol·$g^{-1}$ FW). The non-treated seeds revealed the lowest $H_2O_2$ content of $69.08 \pm 3.69$ mmol·$g^{-1}$ FW (Figure 7a).

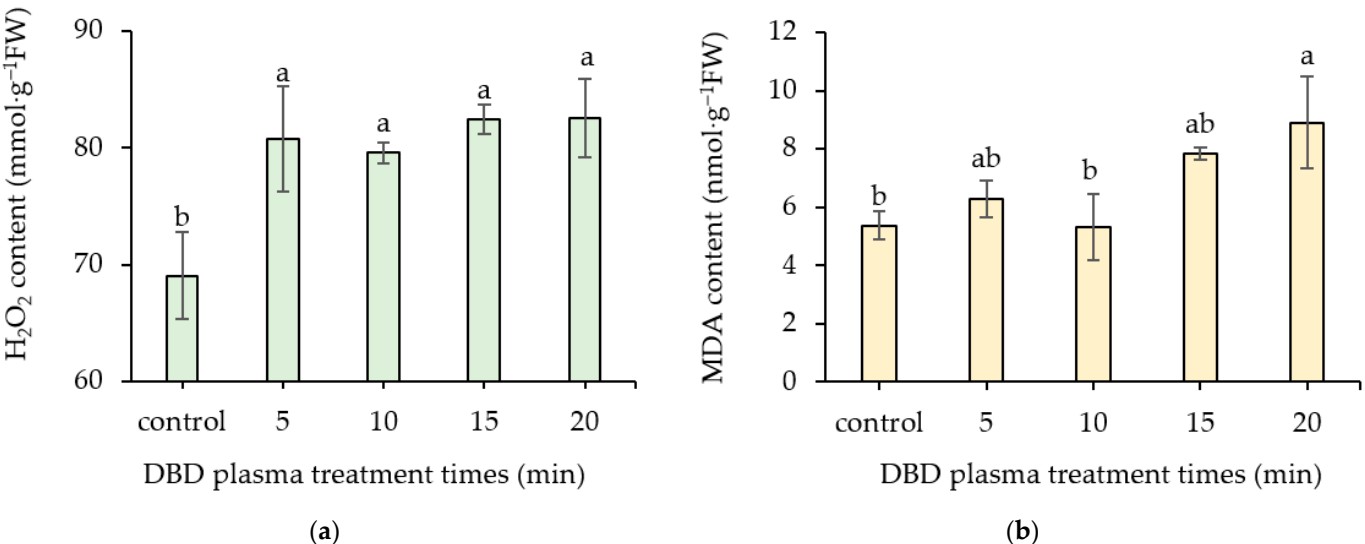

**Figure 7.** Effect of DBD plasma treatments on the contents of $H_2O_2$ (**a**); and MDA (**b**) of kangkong seeds. The data represent means of three replicate samples, and error bars indicate ± SD, with the same lowercase letter indicating a non-significant difference based on Tukey's honestly significant difference test at $p < 0.05$.

### 3.7. Malondialdehyde

The highest MDA content of $8.90 \pm 1.58$ nmol·$g^{-1}$ FW was achieved from the 20 min DBD plasma treatment. However, the MDA concentrations of the control, 5, 10, and 15 min DBD plasma treatments (ranging from $5.38 \pm 0.48$ to $7.84 \pm 0.23$ nmol·$g^{-1}$ FW) were not statistically different (Figure 7b).

### 3.8. Electrical Conductivity

In comparison to the control, the EC values of the DBD plasma treatments significantly increased. The EC values of 5- to 20 min DBD plasma treatments ranged from $75.44 \pm 6.56$ to $86.26 \pm 2.43$ µS·$cm^{-1}$·$g^{-1}$ and were consistently high; however, they did not differ significantly (Figure 8a).

### 3.9. Antioxidant Enzymes

The 5 and 10 min DBD plasma treatments greatly increased the CAT activity (Figure 8b). However, CAT activities of 15 and 20 min DBD plasma treatments did not differ substantially from the control. The APX and SOD activities were not statistically different among the treatments (Figure 8c,d).

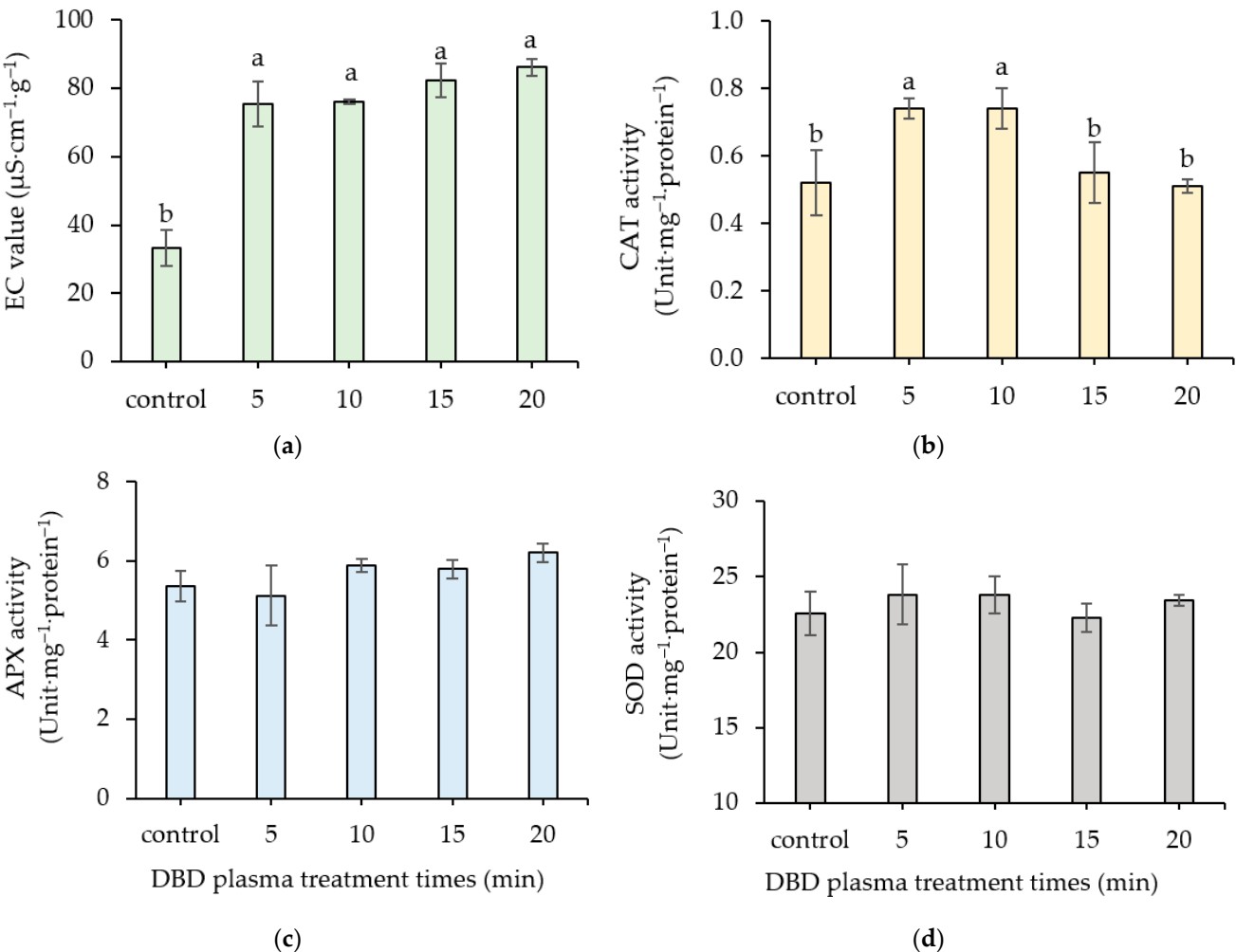

**Figure 8.** Effect of DBD plasma treatments on the EC values (**a**) and the activities of CAT (**b**), APX (**c**), and SOD (**d**) in kangkong seeds. The data represent means of three replicate samples, and error bars indicate ± SD, with the same lowercase letter indicating a non-significant difference based on Tukey's honestly significant difference test at $p < 0.05$.

## 4. Discussion

DBD plasma treatment at appropriate exposure times has been reported to increase seed germination in various species, such as wheat [9,11,37,38], sunflower [30,39], rice [14,16], black gram [10], lentils [37,40], beans [37], spinach [13,15], and radish [18]. Seed germination was induced by plasma-treated seeds due to changes in physical and biochemical factors. In the present study, the SEM images demonstrated significant modifications in the surface of kangkong seeds after being subjected to DBD plasma treatments. Non-treated seeds had relatively smooth surfaces with few fractures. Longer DBD plasma exposure, however, caused the seed surface to significantly deteriorate, showing signs of roughness and wider fissures. These might be due to the numerous energy species, including UV radiation, heat, electrons, positive and negative ions, free radicals, and reactive species, which were released during the plasma discharge [8]. They eroded and bombarded the seed surface; moreover, the degree of cell oxidation was improved by ROS [25]. The changes in seed surface after DBD plasma treatment varied depending on species, plasma device, and treatment duration. For example, the presence of cracks in the barley seed surface widened after being exposed to DBD plasma at 40–80 W for 15 s [41]. The artichoke seed surface showed small holes and cracks; the changes were more serious when DBD plasma treatments were longer, from 3 min to 10 and 15 min [17]. The etching effect on the spinach seed surface under 5 min of DBD plasma treatment was reported [15]. The wax on the black gram

seed coat eroded, resulting in a smoother and thinner seed coat when exposed to longer DBD plasma treatments lasting 90 to 180 s [10]. The seed surface of the radish became significantly rougher after exposure to DBD plasma for 1, 2, and 3 min, particularly with longer exposure times [18].

The changes in seed surface after DBD plasma treatments resulted in a lower water contact angle compared to the non-treated seeds. This occurred due to the increased roughness and widening cracks on the kangkong seed surface after longer DBD plasma treatments, which improved wettability. These results are consistent with those reported by Hosseini et al. [17], who observed similar changes in artichoke seed surfaces following plasma treatment, resulting in a decreased water contact angle and improved wetting properties. In contrast, no structural changes in the seed surface were observed in mimosa [19], pepper, and cucumber seeds [12]. However, the wettability of these seeds still increased due to alterations in the chemical properties of the seed surfaces. Gómez-Ramírez et al. [42] utilized X-ray photoemission spectroscopy to investigate chemical alterations on the surfaces of quinoa seeds after plasma treatments. They found that the outer layers of the plasma-treated seeds exhibited significant oxidation and appeared to be enriched with potassium ions and adsorbed nitrate species.

The water uptake of kangkong seeds strongly increased when the seeds were treated with DBD plasma for 5 to 20 min compared to the control. The water uptake tended to further increase with longer treatment times. This increase in water uptake was attributed to the widening of cracks and higher wettability of the seed coat after DBD plasma treatment. These results were consistent with the findings of several previous studies, including Bormashenko et al. [37], Dobrin et al. [43], da Silva et al. [19], Hosseini et al. [17], and Los et al. [28], which all reported a decrease in the water contact angle after exposing seeds to plasma treatment, resulting in increased water absorption. Furthermore, various studies have indicated that the duration of plasma treatment significantly affects the water uptake of different seeds, such as black gram [10], wheat [43], and *Leucaena* [44]. The changes in the seed surface after plasma treatment improved surface hydrophilicity and enhanced water uptake, thereby promoting germination [28,37].

DBD plasma treatments ranging from 5 to 20 min increased the germination percentage of kangkong seeds compared to the control. However, no significant differences were observed among the various durations of DBD plasma treatments. This might be attributed to the already high germination rates (94–96%) achieved under these treatment conditions. These results were consistent with previous research on sunflower seeds, which showed that after being exposed to DBD plasma for 15 to 240 s, seeds of the cv. Jumbo exhibited a higher germination rate compared to untreated seeds. However, the germination of seeds treated with plasma was not significantly different [39]. In contrast, Han et al. [30] reported that a short 15 s exposure to non-thermal plasma significantly induced germination in sunflower seed, while a longer 30 s exposure had a detrimental effect. Radish seeds exposed to DBD plasma treatment from 1 to 4 min revealed a higher germination percentage than the control, but seed germination decreased at 5 min of plasma treatment [18]. Black gram seeds were subjected to DBD plasma for 20 to 120 s; prolonged plasma exposure resulted in a better germination rate, whereas the germination rate was reduced under the longer treatment of 180 s [10].

$H_2O_2$ is a ROS that can have both beneficial and harmful effects, depending on its concentration [45]. When not properly controlled by antioxidants, it has the potential to adversely affect proteins, lipids, and nucleic acids. Conversely, $H_2O_2$ at an optimal level can promote a number of pathways, including gene expression, protein concentration, oxidative and reductive processes, abscisic acid (ABA) catabolism, and gibberellic acid (GA) biosynthesis [22,25]. Additionally, it plays an important role as a signaling molecule within the seed, promoting seed germination by breaking down seed dormancy [45,46]. In this study, the enhanced germination rate of kangkong seed was associated with the high $H_2O_2$ concentrations in plasma-treated seeds. According to Guo et al. [23], reactive species obtained from plasma treatments penetrated the seeds; however, they had no

specific mechanism until imbibition started [8]. Furthermore, short-lived ROS ($O_2^{\bullet-}$ and $\bullet OH$) may transform into long-lived ROS ($H_2O_2$) during the plasma treatment of dry seeds [22,25]. These demonstrated that $H_2O_2$ from plasma discharge was absorbed by the seeds, exhibiting beneficial effects during germination processes. These results were consistent with wheat [24] and *Arabidopsis* [25] seeds as the increase in $H_2O_2$ contents of plasma-treated seeds resulted in higher germination.

DBD plasma treatments may be related to a decrease in kangkong seed vigor, especially when exposed to a 20 min DBD plasma treatment, which resulted in a significant increase in MDA content. MDA serves as a product of membrane peroxidation and has been utilized as a direct indicator of both lipid peroxidation and membrane damage [27]. Membrane degradation and a subsequent increase in permeability in DBD plasma-treated seeds were observed, as indicated by a significant rise in EC values compared to the control. The low MDA concentration in the 5 and 10 min DBD plasma treatments may be related to a significant rise in CAT activity. Antioxidant enzymes, including APX, CAT, and SOD involved in the oxidative and reductive pathways [22]. These enzymes play an important role in preventing oxidative stresses in plants by scavenging ROS, which controls and limits damage from excessive levels of ROS, and producing biochemicals to enhance germination [8,26]. SOD is recognized as a critical defense mechanism against oxidative damage induced by ROS. It acts as a metalloenzyme, catalyzing the dismutation of superoxide radicals into $H_2O_2$ and oxygen. The activity of APX is linked to the ascorbate-glutathione cycle, as it plays a crucial role in the detoxification of $H_2O_2$ within photosynthetic organisms. CAT serves as a catalyst for the breakdown of $H_2O_2$ into water and oxygen [36]. In the present study, only CAT activity strongly increased after plasma treatments for 5 and 10 min, while APX and SOD activities did not show differential changes. These indicated that CAT had an important role in stress defense in kangkong seeds. These were in accordance with Rahman et al. [24], who reported that the CAT activity of wheat seed increased after plasma treatment, but SOD and APX activity did not change. In 15 s, plasma-treated sunflower seeds showed higher CAT activity, which was related to higher germination than non-treated seeds [30].

## 5. Conclusions

Our results indicated that DBD plasma treatment at different durations stimulated seed germination of kangkong seeds through alterations to the seed surface and the upregulation of $H_2O_2$ content and CAT activity. Additionally, a 5 min DBD plasma treatment appeared to be the most effective for achieving these effects. For future research, we will focus on evaluating the effects of DBD plasma treatments of different durations on the growth and bioactive compounds in kangkong microgreens, as well as the storability of seeds treated with DBD plasma.

**Author Contributions:** Conceptualization, P.R., N.P. and Y.J.; investigation, P.R., P.O., S.S. and K.S.; writing—original draft preparation, P.R. and P.O.; writing—review and editing, P.R., N.P. and Y.J. All authors have read and agreed to the published version of the manuscript.

**Funding:** This research was supported by the Thailand Science Research and Innovation Fundamental Fund, a Ph.D. scholarship from Thammasat University, 1/2022, and the Research Promotion Fund for International and Educational Excellence, 8/2564.

**Data Availability Statement:** The data are contained within the article.

**Acknowledgments:** The authors thank the Faculty of Science and Technology, Thammasat University, and the Center of Advanced Nuclear Technology, Thailand Institute of Nuclear Technology, for providing research facilities.

**Conflicts of Interest:** The authors declare no conflict of interest.

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
