# Peer review of "Germination, Physicochemical Properties, and Antioxidant Enzyme Activities in Kangkong (Ipomoea aquatica Forssk.) Seeds as Affected by Dielectric Barrier Discharge Plasma"

_horticulturae, doi:10.3390/horticulturae9121269_

Round 1
Reviewer 1 Report
Comments and Suggestions for Authors
Ongrak et al., showed that kangkong seeds are affected by DBD plasma. Although there are a few concerns as following. The reviewer feels the data is not appropriate to publish in its present form.
1. The authors mentioned that 5 min was appropriate to treat DBD plasma. However, there was no significant change in germination between 5-20 min. Why don't authors perform the measurements below 5 min?
2. Perform the experiments below 5 min to conclude something. There are many studies where below 5 min treatment given positive results.
3. The introduction is minimal; no highly cited reviews and recently published reviews are included in the manuscript.
4. In the introduction on page 2, the authors mention the Malondialdehyde (MDA), Antioxidant enzymes, etc., but changes in phytohormone, which is one main factor for early germination and seedling growth, were not discussed. However, many studies show a change in phytohormone after DBD treatment on radish seeds, sunflowers, etc.
5. Mention how to calculate the discharge power in section 2.1.
6. In section 2.5, mention how you control the light in the chamber?.
7. In section 2.7, mention how to prepare the standard curve for H2O2 to change in absorbance value to mmol/g FW
8. Few studies show that DBD treatment on seed doesn’t change the seed coat, but there was a surface modification in the present study. So compare with other results.
9. Correct the Figure 6 b Y-axis
10. On page 8, lines 374-376, why did the author add a single reference for some seeds while multiple references for other seeds?. Add more references for all the seeds.
11. On page 10, the authors mentioned, “In this study, the enhanced germination rate of kangkong seed was associated with the high H2O2 concentrations in plasma-treated seeds” but how much H2O2 will be generated in the gas phase? How did this H2O2 concentration increase in seeds after plasma treatment? Enter from outside seed or increase inside in the seeds?
Reviewer 2 Report
Comments and Suggestions for Authors
The paper is well structured, and the subject is novel and of interest. Please attend the following minor recommendations.
Abstract
Add the numbers to the results listed.
Introduction
Line 44- more information of typical germination percentages and rates by traditional methods and be more specific about the negative effects these methods cause.
Lines 69-96 the paragraph gives more information, but it loses focus on the topic, please edit.
M and M
Line 129- fix wording ‘on the’
Results
Fix Figure 6 and 7
Reviewer 3 Report
Comments and Suggestions for Authors
Review comments can be found in the attachment

Round 2
Reviewer 3 Report
Comments and Suggestions for Authors
The author has made good revisions to the manuscript, and I believe it is acceptable and published.